# Loxl3 Promotes Melanoma Progression and Dissemination Influencing Cell Plasticity and Survival

**DOI:** 10.3390/cancers14051200

**Published:** 2022-02-25

**Authors:** Alberto Vázquez-Naharro, José Bustos-Tauler, Alfredo Floristán, Lourdes Yuste, Sara S. Oltra, Antònia Vinyals, Gema Moreno-Bueno, Àngels Fabra, Francisco Portillo, Amparo Cano, Patricia G. Santamaría

**Affiliations:** 1Departamento de Bioquímica UAM, Instituto de Investigaciones Biomédicas Alberto Sols, CSIC-UAM, 28029 Madrid, Spain; avazquez@iib.uam.es (A.V.-N.); jbtauler@gmail.com (J.B.-T.); alfredo.floristan@gmail.com (A.F.); lype_23@hotmail.es (L.Y.); soltra@iib.uam.es (S.S.O.); gmoreno@iib.uam.es (G.M.-B.); fportillo@iib.uam.es (F.P.); 2Instituto de Investigación Sanitaria del Hospital Universitario La Paz-IdiPAZ, 28029 Madrid, Spain; 3Centro de Investigación Biomédica en Red, Área de Cáncer (CIBERONC), Instituto de Salud Carlos III, 28029 Madrid, Spain; 4Oncobell Program, Bellvitge Biomedical Research Institute (IDIBELL), 08908 Barcelona, Spain; avinyals@idibell.cat (A.V.); afabra@idibell.cat (À.F.); 5Fundación MD Anderson Internacional, 28033 Madrid, Spain

**Keywords:** LOXL3, melanoma, melanoma metastasis, genetic mouse model, EMT, cellular plasticity, SNAIL1, PRRX1, phenotype switching

## Abstract

**Simple Summary:**

Malignant melanoma is the most lethal skin cancer due to its aggressive clinical behavior and therapeutic resistance. A comprehensive knowledge of the molecular mechanisms underlying melanoma progression is urgently needed to improve the survival of melanoma patients. Phenotypic plasticity of melanoma cells has emerged as a key process in melanomagenesis and therapy resistance. This phenotypic plasticity is sustained by an epithelial-to-mesenchymal (EMT)-like program that favors multiple intermediate states and allows adaptation to changing microenvironments along melanoma progression. Given the essential role of lysyl oxidase-like 3 (LOXL3) in human melanoma cell survival and its contribution to EMT, we generated mice with conditional melanocyte-specific targeting of *Loxl3*, concomitant to *Braf* activation and *Pten* deletion. Our results supported a key role of Loxl3 for melanoma progression, metastatic dissemination, and genomic stability, and supported its contribution to melanoma phenotypic plasticity by modulating the expression of several EMT transcription factors (EMT-TFs).

**Abstract:**

Malignant melanoma is a highly aggressive tumor causing most skin cancer-related deaths. Understanding the fundamental mechanisms responsible for melanoma progression and therapeutic evasion is still an unmet need for melanoma patients. Progression of skin melanoma and its dissemination to local or distant organs relies on phenotypic plasticity of melanoma cells, orchestrated by EMT-TFs and microphthalmia-associated TF (MITF). Recently, melanoma phenotypic switching has been proposed to uphold context-dependent intermediate cell states benefitting malignancy. LOXL3 (lysyl oxidase-like 3) promotes EMT and has a key role in human melanoma cell survival and maintenance of genomic integrity. To further understand the role of Loxl3 in melanoma, we generated a conditional Loxl3-knockout (KO) melanoma mouse model in the context of BrafV600E-activating mutation and Pten loss. Melanocyte-Loxl3 deletion increased melanoma latency, decreased tumor growth, and reduced lymph node metastatic dissemination. Complementary in vitro and in vivo studies in mouse melanoma cells confirmed Loxl3’s contribution to melanoma progression and metastasis, in part by modulating phenotypic switching through Snail1 and Prrx1 EMT-TFs. Importantly, a novel LOXL3-SNAIL1-PRRX1 axis was identified in human melanoma, plausibly relevant to melanoma cellular plasticity. These data reinforced the value of LOXL3 as a therapeutic target in melanoma.

## 1. Introduction

Malignant melanoma is a highly aggressive tumor that accounts for only about 1% of skin cancers but causes most skin cancer deaths, and its incidence has risen rapidly over the last 50 years [1]. Recent therapeutic advances, mostly provided by immunotherapy and targeted drugs, have not significantly decreased the lethality associated with melanoma, particularly when diagnosed in advanced stages [2]. Melanoma originates in melanocytes, which are melanin-producing cells predominantly located in the skin to protect against DNA damage induced by UV radiation. Melanocyte transformation is due to the accumulation of genetic alterations that ultimately cause uncontrolled proliferation and faulty replicative and DNA repair pathways [3]. Cutaneous melanoma is a highly heterogeneous disease that can be subtyped based on patterns of exposure to UV radiation, clinical and histopathological characteristics, and somatic mutations. These include oncogenic mutations in cell cycle regulators (*CDKN2A*) and members of the MAPK (mainly *NRAS* or *BRAF*) and PI3K/AKT pathways (*PTEN*) [4]. The development of targeted inhibitors against MEK and BRAF represented a milestone in melanoma therapy, but patients eventually relapse due to acquired resistance mechanisms [5]. Immunotherapy drugs, including cytokines and immune checkpoint inhibitors, have emerged recently as first-line treatment, alone or in combined therapies, in most melanoma patients [5,6,7]. However, most patients still present disease progression within 5 years [8].

Despite not being strictly epithelial, neural crest-derived malignancies such as melanoma share similar biological programs with carcinomas that enable tumorigenesis, invasion, and metastasis, such as the epithelial-to-mesenchymal transition (EMT) program, essential during embryogenesis and reactivated in carcinoma metastasis [9,10]. Dissemination of melanoma beyond the primary tumor and local or distant organ colonization relies on an intrinsic plasticity that endows melanoma cells with biological abilities to interact with the changing microenvironment [11]. This phenotypic plasticity is supported by an EMT-like process, regarded as adaptive transcriptional cell states responsible for phenotype switching in response to microenvironment or therapeutic cues [11,12]. Phenotypic plasticity allows the existence of multiple and interchangeable intermediate states, which account for the remarkable ability of melanoma cells to invade and disseminate, and contribute to treatment resistance [12]. This phenotype switching is regulated by EMT-transcription factors (EMT-TFs) such as TWIST1, ZEB1/2, and SNAIL1/2 [13,14]. Microphthalmia-associated TF (MITF), a central regulator of melanocyte differentiation and an oncogene [15], also contributes to cell plasticity. Melanoma cells can switch between different MITF-associated states: MITF^high^, a highly proliferative and differentiated albeit less invasive state; and MITF^low^, which is less proliferative, dedifferentiated, and highly invasive (reviewed in [11,12]). Evidence from in vivo models and human clinical samples has led to a proposed model of phenotype switching during melanoma progression, fine-tuned by ZEB1/2, TWIST1, and SNAIL2, as well as MITF expression [16,17,18,19]. Multiple intermediate states would thus support melanoma progression across different tumor microenvironments, from primary tumors in an invasive phase characterized by MITF^low^, a high ZEB1/ZEB2 ratio, and SNAIL1 expression, to metastatic colonization and outgrowth with MITF^high^, a low ZEB1/ZEB2 ratio, and SNAIL2 expression [12].

LOXL3 (lysyl oxidase-like 3) is a member of the lysyl oxidase family, which comprises prototypical LOX and four LOX-like enzymes (LOXL1-4) that regulate extracellular matrix homeostasis [20,21,22,23]. LOX family members, in particular LOX and LOXL2, are also involved in tumor progression through extracellular and intracellular actions [24,25,26,27]. Different lysyl oxidase genetic mouse models have been instrumental in characterizing their participation in several nontumor pathologies [28,29,30,31,32,33,34]. Regarding cancer, LOX and LOXL2 contribute to breast cancer metastasis [35,36], and LOXL2 is involved in the initiation and progression of head and neck squamous cell carcinoma [34]. Our previous studies showed that intracellular LOXL2 and LOXL3 induced EMT in vitro by Snail1 stabilization [37]. More recently, we unveiled LOXL3’s essential contribution to melanoma cell survival and maintenance of genomic integrity, and showed that LOXL3 is overexpressed in cutaneous melanoma [38]. LOXL3 expression is positively correlated with tumor progression and invasion, and its overexpression is associated with worse prognosis of primary melanoma patients [39].

Given the relevance of LOXL3 in human melanoma, we generated a genetic mouse model to further understand the role of Loxl3 in melanoma formation and progression. For that purpose, we used a mouse model that developed melanoma with 100% penetrance, short latency, and visceral metastases, driven by melanocyte-specific expression of *Braf^V600E^* and inactivation of *Pten* [40]. In this context, deletion of Loxl3 in melanocytes resulted in increased melanoma latency, decreased tumor growth, and notable reduction of lymph node metastatic dissemination. Thus, targeting Loxl3 in vivo extends overall survival. Complementary in vitro and in vivo studies in primary-derived mouse and established human melanoma cell lines confirmed LOXL3 contribution to melanoma progression, in part by modulating melanoma phenotypic plasticity by regulating the expression of SNAIL1 and PRRX1 EMT-TFs. These data reinforced the potential of LOXL3 as a novel therapeutic target in melanoma.

## 2. Materials and Methods

### 2.1. Melanoma Mouse Model and Genotyping

All mouse studies were performed in accordance with protocols approved by the Universidad Autónoma de Madrid Ethics Committee (ref. # CEI-25-587) and the Comunidad de Madrid (PROEX 122/17). Mice were bred in a mixed genetic background (C57BL/6, CD1, and 129v strains). The *Tyr::CreER^T2^; Braf^CA^; Pten^loxP^* strain was purchased from the Jackson Laboratory (Stock No. 013590), and Flippase mice were provided by the Centro Nacional de Investigaciones Oncológicas (CNIO) Transgenic Unit, Madrid, Spain. To generate Loxl3 conditional mice (*Loxl3^loxP^*), embryonic stem cell clones bearing a knockout first allele (*Loxl3tm1a^(EUCOMM)Wtsi^*) were obtained from EUCOMM. This *Loxl3* knockout first allele was generated through the insertion of a cassette after the second *Loxl3* coding exon, containing a splice acceptor site leading to a premature termination codon, followed by a LacZ reporter gene and a neomycin STOP cassette (NeoR), flanked by FRT and loxP sites (Figure 1A). Microinjection of the recombinant clones led to the generation of chimeric animals, and those able to transmit the desired recombination event were used to establish different transgenic strains. Mice heterozygous for the Loxl3 knockout first allele (Loxl3LacZ) were crossed with the Flippase strain in order to obtain a conditional Loxl3loxP allele in which the third Loxl3 coding exon was flanked by loxP sites (Figure 1A). Primers for conventional PCR genotyping are described in Appendix A.

### 2.2. Tamoxifen Treatment

Mouse dorsal skins were shaved and treated with 5 µL of 4-hydroxytamoxifen (4-HT, Sigma-Aldrich, St. Louis, MO, USA, # H7904, 2 mg/mL dissolved in ethanol 100%) during three consecutive days. At the indicated time points, tumors and lymph nodes were collected and processed for histological, immunological, or immunofluorescence analyses.

### 2.3. Tumorigenesis Assays

Control and Loxl3-silenced cells were injected intradermically (5 × 10^3^ in 100 µL of PBS per flank for MeL3 cells or 2 × 10^5^ in 30 µL of PBS for B16-F10 cells) in 8–10-week-old female SCID or C57BL/6 mice, respectively. Before injection, all cells were tested for *Mycoplasma,* and MeL3 cells were also tested for bacterial and fungal contamination as recommended by the Federation of European Laboratory Animal Science Associations (FELASA). Tumor volume was measured twice a week using a caliper and calculated using the following formula: tumor volume = L × W^2^ × (π/6), where L is length and W is width.

### 2.4. Metastasis Assays

B16-F10 (1 × 10^5^) control and Loxl3-silenced cells in 100 µL of PBS were injected in the tail veins of 8–10-week-old female C57BL/6 mice. Lungs were collected and processed for histological analysis 15 days after intravenous injection, and lung metastatic burden was quantified in lung sections.

### 2.5. Histology and Immunohistochemistry

Tissue samples were collected, fixed in 10% formaldehyde, and embedded in paraffin. Tissue sections (5 µm) were stained with hematoxylin and eosin (H/E) solutions or processed for immunohistochemistry performed at the CNIO Histopathology Unit according to the established protocol for Available Techniques (Mouse Samples): Antibodies (https://www.cnio.es/en/research-innovation/services/histopathology/available-techniques/available-techniques-mouse-samples/, accessed on 1 December 2021). The primary antibodies used are listed in Appendix A.

### 2.6. Primary Cell Cultures

Cell lines were derived from transgenic mice bearing 1 cm^2^ melanoma lesions approximately 1 month after 4-HT treatment. Mice were euthanized and sprayed with 70% ethanol twice before dissecting the tumors with a scalpel, and a 1–5 mm^3^ piece was removed from the center of the tumor. The dissected fragment was washed in 70% ethanol for no more than 10 s, and then washed twice with PBS containing 2% penicillin–streptomycin. The tumor was minced into thin pieces with a scalpel blade and resuspended in DMEM:F12 medium (1:1) (Gibco) supplemented with 10% FBS, 2 mM L-glutamine, and 1% penicillin–streptomycin, and transferred into a T25 cell culture flask. Flasks were monitored for the appearance and growth of colonies until near confluence, when cells were then trypsinized and reseeded in T75 flasks for further expansion and evaluation. Stable cell lines were expanded and frozen within 2–3 passages. When needed, cells were thawed and transduced according to the protocol described below, and experiments were performed 3–4 passages after lentiviral infections.

### 2.7. Human Melanoma Cell Culture

The SK-Mel147, WM 1366, and SK-Mel173 melanoma cell lines were generously provided by M.S. Soengas (CNIO, Madrid, Spain). The MeWo, SK-Mel131, M17, and TRP melanoma cell lines were used in a previous study [41]. The WM 1552C, WM 35, WM 793, and WM 115 melanoma cell lines were purchased from ATCC. Cells were cultured in DMEM:F12 medium (1:1) supplemented with 10% fetal bovine serum (Lonza) and maintained at 37 °C in a humidified incubator with 5% CO_2_. The analyses were performed within a few passages after thawing. Cells were routinely tested for *Mycoplasma* contamination.

### 2.8. Mouse Melanoma Cell Lines

The highly metastatic B16-F10 and YUMM1.7 mouse melanoma cell lines were generously provided by Héctor Peinado (CNIO, Madrid, Spain) and Marcus Bosenberg (Yale School of Medicine, New Haven, CT, USA), respectively. They were grown in DMEM and DMEM:F12 medium (1:1), respectively, supplemented with 5% FBS, 2 mM L-glutamine, and 1% penicillin–streptomycin.

### 2.9. Loxl3 Interference

The silencing of *Loxl3* in the mouse melanoma cell lines was achieved by infection with lentiviral particles coding for selected shRNAs (TRCN0000341105, named sh2; and TRCN0000341107, named sh3) from the MISSION TRC shRNA mouse library (Sigma-Aldrich) or the nonmammalian shRNA (nontargeting, NTC) control plasmid (SHC002) in pLKO.1 puro vector. Lentiviruses were packaged in HEK293T cells cotransfecting the lentiviral vector psPAX2 (#12260, Addgene, Watertown, MA, USA) and pMD2.G (#12259, Addgene) with Lipofectamine2000 DNA Transfection Reagent (Thermo Fisher Scientific, Waltham, MA, USA). HEK293T supernatants containing viral particles were harvested 48–72 h after transfection, centrifuged for 5 min at 2500 rpm, and filtered through 0.45 μm membranes. Viral transductions were performed on exponentially growing adherent cultures. Cells grown on 100 mm dishes were transduced with 5 mL of virus-containing supernatants plus 8 μg/mL polybrene. After 8 h, 10 mL of fresh medium was added, and cells were allowed to recover overnight. Two consecutive rounds of infection were performed. Selection with 1–2 µg/mL puromycin began 48 h after the second infection. The efficient deletion of Loxl3 in sh2- and sh3-transduced mouse cell lines was regularly confirmed by qPCR and immunoblot analyses. Primary and transduced cell lines were routinely tested for *Mycoplasma* infection.

### 2.10. Cell Proliferation Assays

For cell proliferation assays, 5 × 10^4^ cells/well were seeded in triplicate in 12-well plates, trypsinized, and counted at the indicated time intervals. For each time point, the total cell number was divided by the number of cells at day 1 after seeding.

### 2.11. Migration Assays

Migration assays of MeL3 cells were performed using modified Boyden chambers with polycarbonate nucleopore membranes (8 μm pore size, Corning BioCoat, Glendale, AZ, USA, 354578). In brief, 5 × 10^4^ cells were seeded in triplicate on the upper part of each chamber, and after incubation for 24 h, nonmigrating cells on the upper surface of the filter were wiped with a cotton swab, and migrated cells on the lower surface of the filter were fixed (4% paraformaldehyde), stained with crystal violet (0.5% in PBS), and counted by examination of five microscopic fields.

### 2.12. Annexin V Staining

Lentiviral-infected cells were seeded at a density of 1 × 10^5^ cells per 6 cm tissue culture plate. After 48–96 h, cells were trypsinized, washed with PBS, incubated with 0.1 μg of propidium iodide and 4 μL of Annexin-V-FITC (ANXVF-200T Immunostep) in binding buffer (5 mM CaCl_2_, 10 mM HEPES pH 7.4, 140 mM NaCl), and analyzed in a BD FACSCanto II Flow Cytometer (BD Biosciences, Franklin Lakes, NJ, USA).

### 2.13. Preparation of Cell Protein Extracts and Immunoblot Analyses

Cultured cells were lysed in radioimmunoprecipitation assay (RIPA) buffer (25 mM Tris-HCl (pH 7.6), 1% Triton X-100, 1% sodium deoxycholate, 0.1% SDS, 150 mM NaCl, 2 mM EDTA, and 2 mM DTT) containing protease and phosphatase inhibitors for 20 min at 4 °C. The extracts were then centrifuged for 20 min at 14,000 rpm, and the supernatants were collected. Equal amounts of total lysates were resolved using SDS-PAGE and analyzed by immunoblot. Membranes were incubated with different primary and secondary antibodies (Appendix A), and detection was carried out with ECL (Perkin-Elmer, Waltham, MA, USA. All Western blot analyses were independently repeated at least three times, and representative results are shown.

### 2.14. Immunofluorescence

Cells grown on coverslips were fixed for 20 min at 37 °C in 4% paraformaldehyde, permeabilized for 5 min with 0.2% Triton X-100, and incubated overnight with the corresponding primary antibodies (Appendix A) diluted in 3% BSA in PBS. Incubation with appropriate secondary antibodies (Appendix A) proceeded for 1 h. Slides were mounted with Prolong (Thermo Fisher Scientific) and analyzed using an inverted Zeiss LSM710 confocal microscope with a 40×/1.40 Plan-Apochromatic objective. Images shown are single sections of a z-series, and pictures were processed with ZEN 2009 Light Edition (Carl Zeiss MicroImaging, Jena, Germany).

### 2.15. DNA Foci Quantification

For quantification of DNA foci, image analysis was performed with ImageJ software as described previously [42]. Briefly, nuclei were selected by DAPI-positive staining, and foci displaying a γH2AX/53BP1 signal above a certain threshold, established as background, were quantified. A minimum of 100 cells were analyzed per experiment and condition.

### 2.16. RNA Isolation, Reverse Transcription, and qPCR Analysis

Total RNA from mouse cultured cells was extracted using TRIzol (Thermo Fisher Scientific, Waltham, MA, USA) and GeneJET RNA Purification Kit (Thermo Fisher Scientific) spin columns. On-column DNase treatment was performed according to standard procedures to avoid the presence of residual DNA. For cDNA synthesis, 2 μg of RNA was retrotranscribed with M-MLV RT (Promega Corporation, Madison, WI, USA). Quantitative PCR (qPCR) analyses were performed in a 7900 HT FAST Real-Time PCR System thermocycler (Thermo Fisher Scientific) using *Power* SYBR Green dye (Thermo Fisher Scientific). Reverse transcription from human melanoma cell lines was performed with the First Strand cDNA Synthesis Kit (Life Technologies, Carlsbad, CA, USA) using random hexamer primers. The qPCR was performed in a LC 480 machine using SYBR Green Mastermix (Roche Life Science, Penzberg, Germany). Primer pairs used for the qPCR analysis are listed in Appendix A. Relative quantification of gene expression was normalized to *GAPDH* or *L32* levels and calculated using a 2-(ddCt) ± S.D. formula.

### 2.17. Human Melanoma Data Mining and Analysis

To assess *LOXL3*, *SNAI1,* and *PRRX1* gene expression in skin cutaneous melanoma patients, an in silico study was performed using data from The Cancer Genome Atlas (TCGA) cohort dataset (*n* = 469) [43]. Correlations between paired genes were assessed by Pearson correlation. Patients samples were subdivided in *LOXL3* expression of high (higher than *LOXL3* mean expression, *n* = 287) or low (lower than *LOXL3* mean expression, *n* = 182), and *SNAI1* and *PRRX1* expression levels in these groups were analyzed by *t*-tests. *LOXL3*, *SNAI1,* and *PRRX1* expression were assessed in patients defined according to the *BRAF* mutational status (mutated or WT) by *t*-tests. Results were considered significant with a *p*-value < 0.05. Statistical analyses were performed using R Bioconductor (https://www.bioconductor.org, accessed on 25 November 2021).

### 2.18. Statistical Analyses

Unless otherwise indicated, numerical data are expressed as mean  ±  SEM. No statistical methods were used to predetermine the sample/group sizes. Sample sizes, number of replicates, and normalization methods are indicated in each figure legend. Statistical analyses were performed using GraphPad Prism 8.0 software, and the corresponding method is indicated in each figure legend. The statistical significance of differences between groups is indicated by a number or asterisks (*, 0.01 < *p* < 0.05; **, 0.001 < *p* < 0.01; ***, *p* < 0.001).

## 3. Results

### 3.1. Generation and Characterization of a Loxl3 Conditional Melanoma Mouse Model

To obtain a melanoma mouse model with selective Loxl3 loss of function, conditional knockout mice for *Loxl3* (*Loxl3^loxP^*) were first generated as described in the Materials and Methods section and Figure 1A. *Loxl3^loxP^* mice were then crossed with *Tyr::CreER^T2^; Braf^CA^; Pten^loxP^* mice [40], and their offspring were intercrossed until appropriate control (*Braf Pten L3^wt^*) and experimental (*Braf Pten L3^loxP^*) animals were obtained and confirmed by diagnostic PCR (Figure 1B). In these mice, administration of 4-HT resulted in the activation of the conditionally active CreERT2, which was expressed specifically in melanocytes driving the expression of constitutively active *Braf^V600E^* and concomitant inactivation of *Pten* and *Loxl3* gene expression (Figure 1C).

### 3.2. Deletion of Loxl3 Increases Latency and Reduces Melanoma Tumor Growth and Overall Mice Survival

The administration of 4-HT topically on the back skins of *Braf Pten L3^wt^* and *Braf Pten L3^loxP^* mice (days 1–3 following weaning) led to the appearance of highly pigmented lesions that were detected within 20–30 days (Figure 2A). The lesions (Loxl3 WT or Loxl3 KO) appeared mostly on regions to which 4-HT was applied, as previously described [40], and 4-HT-driven genomic rearrangements were confirmed by PCR on relevant tissues (Figure 1B). All the *Braf Pten* animals that were subjected to 4-HT treatment developed pigmented lesions that eventually progressed to melanomas with different kinetics depending on their *Loxl3* genotype (Figure 2A). Histological analysis of the 4-HT-induced primary lesions in *Braf Pten* mice showed their location at the dermal–epidermal junction (Figure 2B), and their melanocytic origin was confirmed by the immunodetection of Tyrosinase-related protein 2 (Tyrp2) (Figure 2C), an enzyme involved in the synthesis of melanin.

We monitored 4-HT-treated *Braf Pten* mice over time to determine the rate and progression of Loxl3 WT and KO pigmented lesions. Specific deletion of *Loxl3* in the melanocytes and tumors formed in *Braf Pten L3^loxP^* mice delayed the incidence, development, and growth of pigmented lesions compared to control *Braf Pten L3^wt^* animals (Figure 2A,B,E). The number and spread of pigmented melanoma cells found in the dermis of 4-HT treated *Braf Pten L3^loxP^* mice was decreased compared to control animals (Figure 2A,B,E). Correspondingly, overall mice survival was significantly increased in the absence of Loxl3 (Figure 2F), confirming the role of Loxl3 in melanocyte biology and transformation. The inactivation of Loxl3 extended the median disease latency by 32% (from 14 to 21 days; HR  =  0.68, CI  =  0.4 to 1.16) (Figure 2D) and the overall survival time by 51% (HR  =  0.49, 95% CI  =  0.28 to 0.85) (Figure 2F) of *Braf Pten* mice. These facts, together with the decrease in tumor growth in the Loxl3 KO animals (Figure 2E), provided the first genetic evidence supporting the involvement of Loxl3 in melanoma development in vivo.

### 3.3. Selective Inactivation of Loxl3 in the Melanocytes Decreases Metastatic Dissemination

Upon examination of the mice treated with 4-HT, evidence of melanoma spread was clearly seen in lymph nodes (Figure 3A) but not in lungs or spleens, metastatic sites previously reported when 4-HT was administered systemically [40,44]. Macroscopically, proximal (subiliac) and distant (superficial parotid) lymph nodes were remarkably smaller and showed less pigmentation in mice bearing Loxl3 KO melanomas than in Loxl3 WT tumor animals 42 days upon 4-HT-induced development of melanomas (Figure 3A). Indeed, distant lymph nodes from Loxl3 KO mice resembled those from 4-HT untreated animals that did not develop tumors. To assess the melanocytic origin of the metastatic lesions found, lymph nodes were stained with the melanin enzyme Tyrp2 and with Sox10, a key transcription factor responsible for the establishment and maintenance of the melanocytic lineage [45] and a reliable marker for the detection of metastatic melanoma in sentinel lymph nodes [46] (Figure 3B). Loxl3 deletion notably inhibited the appearance of metastatic lesions in lymph nodes of animals treated with 4-HT as determined by independent Tyrp2 staining (Figure 3C). In particular, metastatic lesions in distant lymph nodes were detected in only 4% of mice bearing Loxl3 KO melanomas (1 out of 28 mice) while present in 81% of mice bearing Loxl3 WT melanomas (25 out of 31) (Figure 3C). The assessment of the metastatic incidence in local and distant lymph nodes from both mice cohorts and the overall quantification of the Tyrp2-stained area (Figure 3C,D) indicated a role for Loxl3 in melanoma dissemination, independently of Loxl3 involvement in primary tumor growth. This fact was supported by the remarkable inhibition of metastasis incidence detected upon Loxl3 inactivation, likely uncoupled from delayed tumorigenesis.

### 3.4. Loxl3 Silencing Is Detrimental to Melanoma Cell Growth

To understand the molecular mechanisms by which Loxl3 contributes to mouse melanomagenesis, we derived primary cell lines from our *Loxl3* melanoma mouse model as detailed in the Materials and Methods section and Figure 4A. Several cell lines were established from melanomas grown on Loxl3 WT mice, while no single cell line was derived from those grown on Loxl3 KO animals, despite numerous attempts. The molecular characterization of the established cell lines demonstrated that only one cell line, named MeL3, recapitulated the expected protein expression pattern; i.e., the presence of melanocytic lineage markers Tyrp2 and Sox10, enhanced ERK phosphorylation due to Braf activation, and Pten loss, while most melanoma-derived cell lines, such as 137d, did not (Figure 4B). ERK activation, Pten loss, and Tyrp2 expression also were seen in the YUMM1.7 melanoma cell line derived from induced tumors in the *Braf Pten* mouse model [47], whereas Sox10 and Tyrp2 were expressed by human A375P cells, representative of primary cutaneous human melanoma cell lines (Figure 4B). MeL3 cells displayed a mesenchymal and amelanotic phenotype, common in melanoma-derived cell lines (Figure 4C). Since we were not able to derive any Loxl3 KO cell line, we reasoned that Loxl3 is necessary for melanoma cell survival in 2D growth conditions, as we have previously shown for human melanoma cells [38]. We then silenced Loxl3 expression in the MeL3 cell line by lentiviral transduction using two specific shRNAs against Loxl3 (sh2 and sh3) and a nontargeting control (NTC) sequence. Although Loxl3 silencing was achieved and maintained over time by both shRNAs without changing the mesenchymal phenotype (Figure 4C) or altering the expression of MeL3 distinctive markers (Figure 4D), the depletion of Loxl3 significantly reduced cell proliferation (Figure 4E). However, Loxl3 silencing did not induce cell apoptosis (Figure 4F), as is the case in human melanoma cells [38], but promoted DNA damage, determined by the accumulation of γH2AX and 53BP1 foci in Loxl3-depleted MeL3 cells compared to control cells (Figure 4G,H). Resembling the human setting, Loxl3 supported genome stability in mouse melanoma-derived cells, and the accumulation of DNA damage upon Loxl3 deletion would thus contribute to impaired tumor cell proliferation.

Altogether, these results confirmed that Loxl3 is required for melanoma cell growth both in vivo and in vitro.

### 3.5. Loxl3 Expression in Melanoma Cells Contributes to Tumor Progression

In order to analyze whether the contribution of Loxl3 to melanomagenesis is cell-autonomous, we injected MeL3 control and Loxl3-silenced cells in the dorsal flanks of immunodeficient mice. Upon injection, melanoma-derived cells depleted for Loxl3 formed a lower number of tumors than control cells, and the tumors developed from Loxl3-silenced cells showed increased latency and delayed growth compared to those established from control cells (Figure 5A). Moreover, the volume of the tumors formed by MeL3 cells depleted for Loxl3 at the endpoint was considerably smaller than the volume of those formed by control cells (Figure 5B). Since Loxl3 is required for the migration of MeL3 melanoma cells in vitro (Figure 5C), we closely examined all mice injected with MeL3 cells at the endpoint. No differences in size or swelling were seen in the lymph nodes from mice with control and Loxl3-depleted xenografts. In addition, no metastatic foci were detected in the lymph nodes or lungs of any mice, which could be explained by the fast primary tumor growth of control MeL3 xenografts that reached 1 cm^3^ in just three weeks, precluding evident metastatic dissemination.

To further assess the involvement of Loxl3 in melanoma progression in an immunoproficient setting, we used the highly metastatic B16-F10 mouse melanoma cell line, which can be transplanted into syngeneic C57BL/6 mice [48]. Loxl3 silencing in these cells reduced cell proliferation significantly upon transduction with sh3 (Figure 6A,B). Accordingly, primary tumors developed in the flanks of syngeneic mice showed a delayed growth upon inoculation of Loxl3-silenced B16-F10 cells compared to those derived from control cells, accompanied by a reduction in tumor incidence and an increase in tumor latency (Figure 6C). We suspected that the highly proliferative ability of B16-F10 cells conceals a stronger impact of Loxl3 depletion, as seen in our previous models. Nonetheless, a clear accumulation of DNA damage was also detected in B16-F10 Loxl3-silenced cells compared to control ones (Figure 6D,E), which could explain their reduced proliferative and tumorigenic abilities, and could support Loxl3’s role in maintaining genomic stability. We indeed observed a diminished ability for clonogenic survival and transformation of B16-F10 cells upon Loxl3 depletion, as determined by anchorage-dependent and -independent colony formation assays, respectively (Appendix A). Compelled by this result, and given the lung colonization ability of B16-F10 cells, we inoculated control and Loxl3-silenced B16-F10 cells in the tail veins of syngeneic mice. De visu analyses and histological characterization of the recovered lungs showed a marked reduction in the lung metastatic burden of Loxl3-silenced cells compared to controls (Figure 6F,G).

Collectively, the results depicted thus far, using different melanoma cell lines and both immunodeficient and immunoproficient mice, indicated that Loxl3 contributes to mouse melanomagenesis and metastatic dissemination by tumor-cell-autonomous mechanisms.

### 3.6. Loxl3 Is Involved in Melanoma Cell Plasticity

Our group unveiled that human LOXL3 was able to interact with Snail1 and repress E-cadherin gene expression, behaving as an EMT-TF [37]. Therefore, we hypothesized whether Loxl3 might be involved in mouse melanoma cell plasticity underlying Loxl3’s contribution to tumor progression and metastasis. To avoid confounding effects due to intrinsic cell heterogeneity within tumor samples, we analyzed the expression levels of *Mitf*, a key TF governing the state of melanocyte differentiation [49], the levels of which are related to the proliferative and invasive states of melanoma (reviewed in [11]), and a panel of EMT-TFs in the MeL3 cell line with or without Loxl3 expression. In this setting, we detected a rise in *Mitf* expression upon Loxl3 silencing (Figure 7A). Even though the levels of the EMT-TFs *Zeb1*, *Zeb2,* and *Snai2* did not change, *Twist1* expression was clearly upregulated, and a remarkable reduction in *Snai1* and *Prrx1* levels was detected in Loxl3-silenced cells compared to MeL3 control cells (Figure 7A). At the protein level, we were able to confirm the significant downregulation of Snail1 and Prrx1 upon Loxl3 silencing, suggesting that Loxl3 might control Prrx1 and Snail1 expression in Braf^V600E^-mutated mouse melanoma cells (Figure 7B,C). Indeed, when we analyzed the expression of these genes in a panel of human melanoma cell lines, the majority of cells carrying activated BRAF^V600E^ mutation expressed LOXL3, SNAIL1, and PRRX1 as determined by gene and protein analyses (Figure 7D,E). In addition, in 469 samples of skin cutaneous melanomas from the TCGA [43], a positive correlation was found for *LOXL3* gene expression with both *SNAI1* and *PRRX1* expression (R = 0.4 and R = 0.45, respectively), as well as for *PRRX1* with *SNAI1* expression (R = 0.45) (Figure 7F). Remarkably, a significant upregulation of *SNAI1* and *PRRX1* mRNA was seen in tumors displaying high levels of *LOXL3* expression (Figure 7G), whereas *LOXL3*, *SNAI1,* and *PRRX1* were notably upregulated in *BRAF*-mutated tumors (Figure 7H). These results indicated that, in melanoma, and particularly in a BRAF oncogenic environment, the LOXL3-SNAIL1-PRRX1 axis might favor a transition toward a malignant phenotype, activating invasion and migration.

Altogether, these results confirmed that Loxl3 is required for melanoma cell growth both in vivo and in vitro.

## 4. Discussion

Deregulated LOXL3 has been linked to several connective tissue disorders [29,30,50,51,52,53,54], consistent with the well-established role of lysyl oxidases as extracellular matrix enzymes contributing to tissue homeostasis [20,23]. Regarding cancer, few reports have involved LOXL3 in human cancer [55,56,57,58]. Importantly, our previous studies unveiled a critical role for LOXL3 in melanomagenesis, and established that melanoma cells are addicted to LOXL3 expression. Mechanistically, we showed that LOXL3 is responsible for maintaining melanoma genome stability, since its depletion results in impaired DNA damage response and defective mitotic completion, leading to apoptotic cell death. We found that LOXL3 is expressed in an ample cohort of primary and malignant human cell lines and uncovered an association of LOXL3 expression with tumor samples carrying melanoma driver mutations [38]. LOXL3 relevance in melanoma is further supported by a recent analysis of 373 biopsies that correlates LOXL3 expression with tumor progression, invasion, and worse prognosis of primary melanoma patients [39]. Nonetheless, the in vivo role of endogenous LOXL3 in melanomagenesis has not been previously explored. Genetic mouse models provide crucial understanding of cancer initiation and progression; in particular, the genetic model used in this study, carrying activated Braf and Pten deletion, recapitulates relevant pathophysiological features of human melanoma, including melanocyte transformation, local invasion, and metastasis, closely resembling the clinical course of the disease [40]. Deletion of Loxl3 and simultaneous activation of Braf and loss of Pten in melanocytes increases melanoma latency and decreases tumor growth, resulting in increased overall survival of mice lacking Loxl3 expression. Since Loxl3 depletion is restricted to melanocytes, these results confirmed that LOXL3 cooperates with oncogenic BRAF to promote melanomagenesis, as we demonstrated in in vitro experiments in which the overexpression of LOXL3 in immortalized human melanocytes expressing BRAF^V600E^ led to their malignant transformation [38].

In contrast to human melanoma cell lines in which we found that *LOXL3* knockdown resulted in cell death [38], *Loxl3* silencing in mouse melanoma cells reduced cell proliferation without promoting apoptosis. Even if an increased accumulation of DNA damage was seen in the absence of Loxl3 in these cell lines, no cell cycle abnormalities were detected, indicating that mouse melanoma cells were able to survive without Loxl3 in the long term, either because the accumulation of DNA damage was lower than in the human setting, or mouse cells could activate a somewhat functional DNA damage response that prevented additional genomic aberrations in established cell cultures. However, primary cell lines deleted for *Loxl3* could not be established from Loxl3 KO tumors, indicating an essential requirement of Loxl3 to sustain initial growth in 2D cultures. On the other hand, mouse Loxl3 supported melanoma cell migration abilities in vitro and tumor growth in vivo both in MeL3 cells and in our melanoma mouse model, similar to human LOXL3. Importantly, Loxl3 is required for melanoma metastasis, both lymphatic, determined by the dramatic decrease of lymph node invasion in Loxl3 KO mice, and visceral, supported by reduced lung metastatic burden in experimental metastasis assays in syngeneic mice upon tail vein injection of B16-F10 cells silenced for Loxl3. Altogether, these results corroborated that mammalian Loxl3 is involved in melanomagenesis, contributing to the malignant transformation of melanocytes and melanoma metastatic competence.

To understand the molecular mechanisms responsible for the critical role of mouse Loxl3, we explored Loxl3 involvement in melanoma cell plasticity, given our previous data linking LOXL3 to EMT [37]. Indeed, primary melanoma cells showed an upregulation of *Mitf* and *Twist1* and a downregulation of *Snai1* and *Prrx1* levels upon *Loxl3* silencing. Melanoma phenotype switching has been proposed to be responsible for melanoma heterogeneity, controlled by cell-autonomous and microenvironment factors that impinge on MITF and EMT-TF regulation, particularly on ZEB1/2 and SNAIL1/2 [11]. Our results suggested that MeL3 primary-derived melanoma cells are in an invasive MITF^low^ state associated with Loxl3 expression, which switches to a proliferative and low migratory MITF^high^ state upon Loxl3 depletion while maintaining an undifferentiated cell phenotype suggestive of an intermediate state [11]. This was supported in vivo by the fact that Loxl3-depleted melanoma cells originated less-proliferative (MeL3) and invasive (B16-F10) tumors, which was recapitulated in the Loxl3 KO mouse melanoma model. In the absence of the tumor microenvironment, Zeb1/2 levels were unaffected by Loxl3 silencing in MeL3 cells, although both gene and protein levels of Snail1 and Prrx1 EMT-TFs were markedly downregulated. In carcinomas, SNAIL1 represents a strong epithelial repressor that cells upregulate to detach from their neighbors [59]. PRRX1, a strong mesenchymal inducer, is activated to promote the acquisition and maintenance of robust mesenchymal features for tumor cells to migrate to their destination [60]. Multiple studies supported the role of both EMT-TFs in carcinoma progression and metastatic dissemination by favoring a mesenchymal/invasive phenotype [61,62], although their specific contribution to melanomagenesis was not clearly defined. Snail1-induced EMT was involved in melanoma metastasis by facilitating immune evasion [63], while there is limited data linking PRRX1 to melanoma [60,64]. The association of Loxl3 expression with Snail1 and Prrx1 levels seen in primary melanoma-derived cells is reproduced in a collection of established human melanoma cell lines. Most cell lines expressed LOXL3 independently of their mutational status, as we previously described [38], while BRAF^V600E^-mutated melanoma cells expressed elevated levels of LOXL3, SNAIL1, and PRRX1, the genes for which were significantly upregulated in *BRAF*-mutated patients´ samples. Notably, high *LOXL3* and *SNAI1* or *PRRX1* mRNA levels were significantly associated in human cutaneous melanoma patients, unveiling a previously unknown LOXL3–SNAIL1–PRRX1 axis relevant to melanoma biology.

The results presented here suggested that the LOXL3–SNAIL1–PRRX1 axis contributes to phenotypic switching in melanoma and supports tumor progression. Further experiments are needed in order to understand the mechanisms governing the regulation of LOXL3, SNAIL1, and PRRX1 factors in melanomagenesis and their specific contribution to melanoma cell plasticity. In addition, it will be interesting to explore whether the role of LOXL3 in extracellular matrix remodeling also contributes to the malignant progression of melanoma, which might also impact melanoma cell plasticity, promoting metastasis and the emergence of drug resistance.

This study builds upon our previous data involving LOXL3 in human melanomagenesis, and offers additional distinct and substantial results that warrant the exploration of LOXL3 as a therapeutic target in malignant melanoma, a very aggressive condition in need of improved treatment options.

## 5. Conclusions

In summary, our data confirmed the critical action of LOXL3 in promoting melanoma tumor growth and tumor invasion. Given the prevalence of *BRAF* and *PTEN* alterations in human melanoma, our results, which demonstrated that in vivo targeting of Loxl3 was deleterious to melanomagenesis in a mouse model harboring activated Braf^V600E^ and loss of Pten, warrant future strategies aimed at blocking cellular LOXL3. We have exposed a novel LOXL3–SNAIL1–PRRX1 axis relevant to melanoma phenotype switching, the targeting of which might represent an opportunity to combat resistance to targeted and immune-based therapies, a remaining challenge in the clinical management of melanoma.

## Figures and Tables

**Figure 1 cancers-14-01200-f001:**
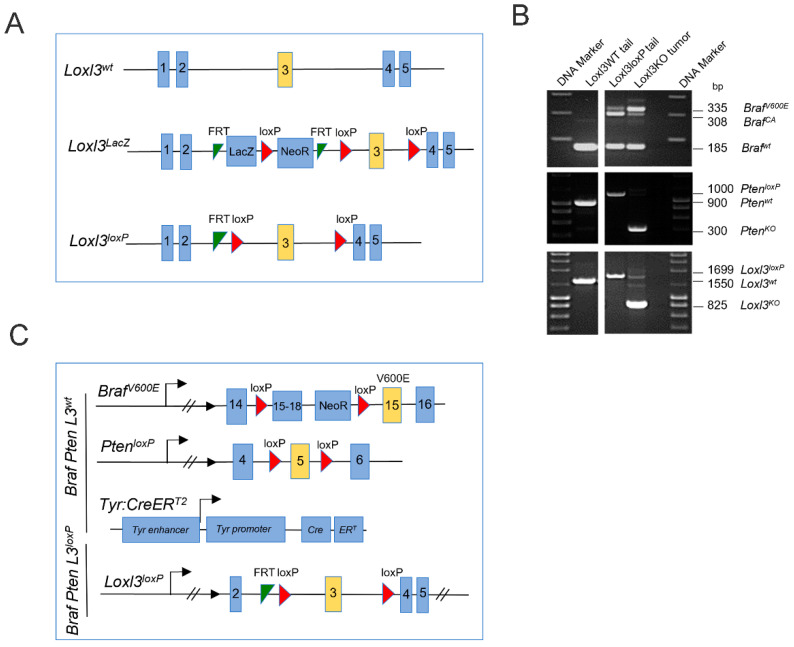
Strategy to generate mice with conditional melanocyte-specific targeting of *Loxl3*. (**A**) Schematic representation of the Loxl3 alleles present in the mice studied: wild-type *Loxl3^wt^* allele, knockout first allele *Loxl3^LacZ^* modified through Flippase to obtain a conditional *Loxl3^loxP^* allele. See Materials and Methods for details. (**B**) Confirmation by diagnostic PCR of the genetic changes produced by CreERT2 recombinase in *Braf*, *Pten,* and *Loxl3* genes in genomic DNA from tails from *Braf^+/+^ Pten^+/+^ Loxl3^+/+^* WT untreated mice (left panels), and tails and tumors from conditional *Braf^V600E^ Pten^loxP^ Loxl3^loxP^* mice upon 4-HT treatment (right panels). The expected size in base pairs (bp) of the different PCR products from the corresponding alleles is indicated before each of them on the right side. (**C**) Schematic representation of the *Braf^V600E^*-, *Pten^loxP^*-, *Tyr:CreER^T2^*-*,* and *Loxl3^loxP^*-modified alleles included in the conditional mouse model. Allele nomenclature: *Braf^V600E^*: *Braf* allele targeted with *Braf^V600E^* modification; *Braf^CA^*: constitutively active *Braf* allele upon CreERT2-mediated recombination; *Pten^loxP^*: *Pten* allele modified to include loxP-flanked exon 5; *Pten^KO^*: loss of function *Pten* allele upon CreERT2-mediated recombination; *Tyr:CreER^T2^*: 4-HT conditionally active CreERT2 expressed under the control of the *Tyrosinase* (Tyr) promoter; *Loxl3^loxP^*: *Loxl3* allele including loxP-flanked exon 3 after Flippase-induced recombination of *Loxl3^lacZ^* allele; *Loxl3^KO^*: loss of function *Loxl3* allele upon CreERT2-mediated recombination; wt denotes wild-type alleles. FRT: Flippase recognition sites; loxP: Cre recombinase recognition sites.

**Figure 2 cancers-14-01200-f002:**
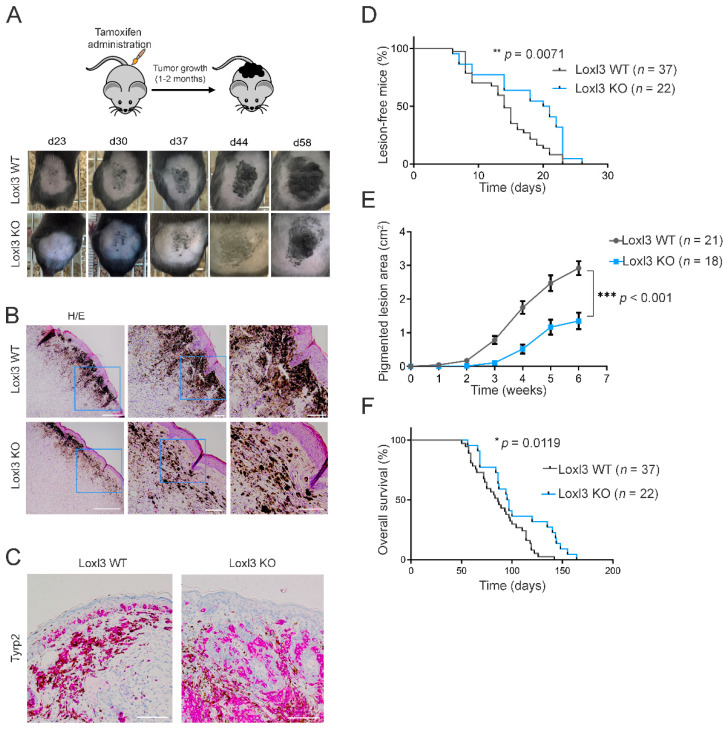
*Loxl3* targeting in a melanoma mouse model delays tumor development and progression. (**A**) Schematic representation of the protocol followed to induce melanoma development on the backs of *Braf Pten L3^wt^* and *Braf Pten L3^loxP^* mice. Representative pictures from the appearance and evolution of the lesions at indicated days upon tamoxifen treatment in *Braf Pten L3^wt^* (Loxl3 WT) and *Braf Pten L3^loxP^* (Loxl3 KO) mice are shown (lower panels). (**B**) Hematoxylin and eosin (H/E) staining of skin sections from Loxl3 WT and Loxl3 KO lesions. Blue frames denote the amplified areas shown in the middle and right side images. Scale bars: 500 µm (left pictures), 100 µm (middle and right pictures). (**C**) Tyrp2 staining by IHC of skin sections from Loxl3 WT and Loxl3 KO lesions. Scale bar: 100 µm. (**D**) Percentage (%) of animals free of lesions from *Braf Pten L3^wt^* (Loxl3 WT) and *Braf Pten L3^loxP^* (Loxl3 KO) backgrounds at different days upon 4-HT treatment. (**E**) Area of pigmented lesions in *Braf Pten L3^wt^* (Loxl3 WT) and *Braf Pten L3^loxP^* (Loxl3 KO) mice at different weeks upon 4-HT administration. (**F**) Kaplan–Meier survival curve of *Braf Pten L3^wt^* (Loxl3 WT) and *Braf Pten L3^loxP^* (Loxl3 KO) mice following 4-HT treatment. The *p*-values were calculated by Mantel–Cox test (**D**,**F**) and two-sided unpaired Student’s *t*-test (**E**); *n* indicates the number of monitored animals with the indicated genotype (**D**–**F**). * *p* < 0.05, ** *p* < 0.01, and *** *p* < 0.001.

**Figure 3 cancers-14-01200-f003:**
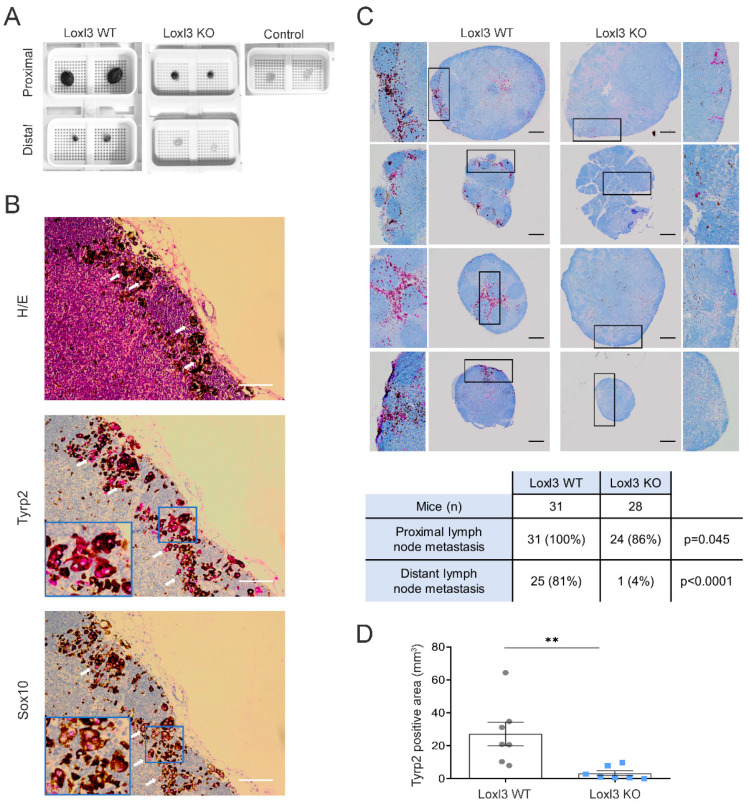
Loxl3 inactivation in melanocytes decreases melanoma lymphatic dissemination. (**A**) Representative images of proximal and distal lymph nodes collected at 6 weeks after 4-HT treatment from Loxl3 WT and Loxl3 KO mice in the *Braf Pten* genetic background. The image on the upper right shows lymph nodes collected from untreated age-matched control *Braf Pten L3^wt^* animals. (**B**) H/E, Tyrp2, and Sox10 stainings of lymph node sections from *Braf Pten L3^wt^* mice bearing 4-HT-induced melanomas. Blue frames indicate an amplified area from the middle sections depicted in the lower left quarters (middle and lower panels). White arrows point to cytoplasmic (Tyrp2) and nuclear (Sox10) detection. Scale bars: 100 µm. (**C**) Representative images (top) of Tyrp2 staining in proximal and distal lymph node sections from Loxl3 WT and proximal lymph node sections from Loxl3 KO animals bearing melanomas developed upon 4-HT administration. Black frames denote amplified sections shown on the left (Loxl3 WT) and right (Loxl3 WT) sides; some of them are rotated 90 °C regarding the original image. Scale bars: 200 µm. The chart (bottom) indicates the number and percentage of Tyrp2-positive distal and proximal lymph nodes detected in Loxl3 WT and KO mice sacrificed at 42 days after 4-HT treatment. (**D**) Quantification of total Tyrp2 stained area in lymph node sections collected from mice as detailed in (**A**). Lymph nodes from Loxl3 WT and KO mice (*n* = 7) were analyzed. The *p*-value was calculated by two-sided unpaired Student’s *t*-test. ** *p* < 0.01.

**Figure 4 cancers-14-01200-f004:**
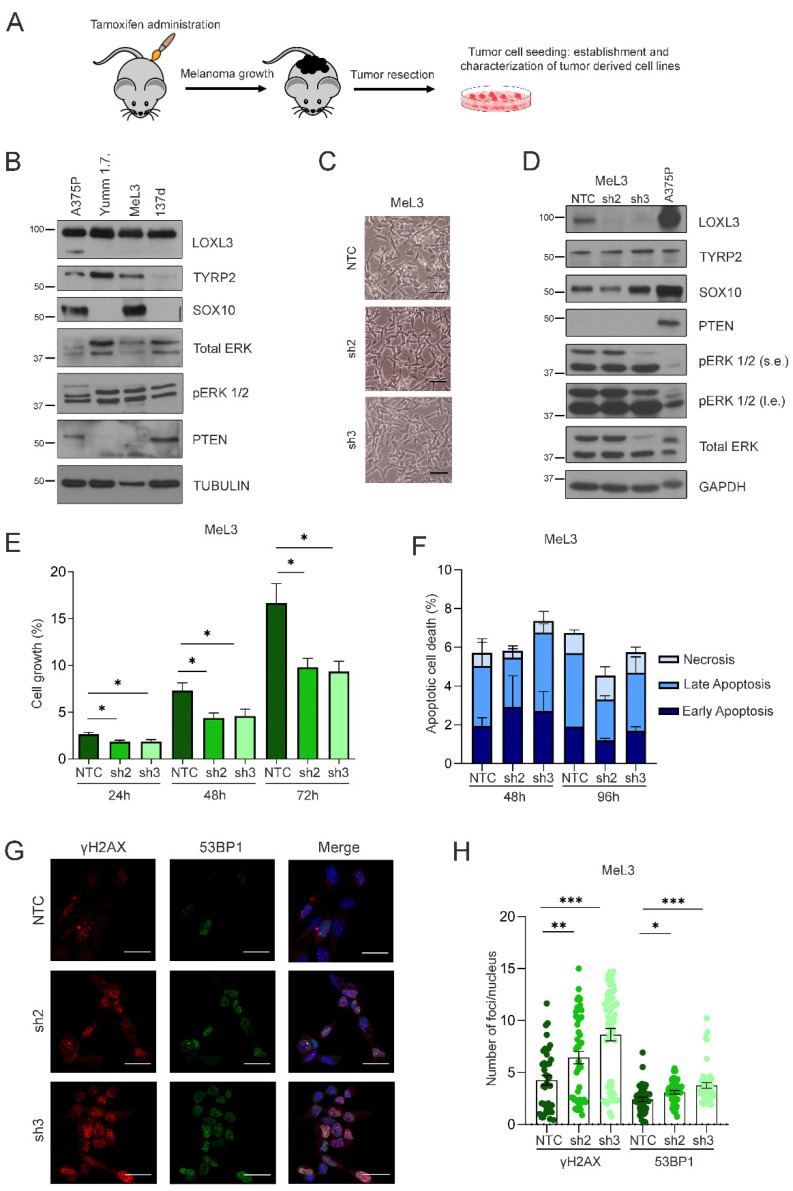
Loxl3 silencing in primary melanoma-derived cells decreases proliferation and increases DNA damage. (**A**) Scheme showing the derivation and establishment of primary cell lines from Loxl3 WT and KO melanomas. (**B**) Characterization by protein expression analyses of melanoma-derived MeL3 cell line compared to human (A375P) and mouse (YUMM1.7) established melanoma cell lines. The 137d line represents a negative control of primary tumor-derived cells without the expected protein pattern regarding TYRP2, SOX10, pERK1/2, total ERK, and PTEN protein expression. Tubulin was used as a loading control. Note that the LOXL3Δ isoform described in human cell lines [38] is not detected in mouse cell lines. Uncropped Western Blots and densitometry analyses can be found at Appendix A. (**C**) Phase-contrast images of MeL3 cells upon Loxl3 silencing using two specific shRNAs (sh2 and sh3) and a nontargeting control (NTC) sequence. Scale bars: 100 µm. (**D**) Immunoblot analyses of melanoma markers (TYRP2, SOX10) and PTEN and BRAF status in control (NTC) and Loxl3-silenced (sh2 and sh3) MeL3 cells. Human.A375P melanoma cells were included as an additional control. GAPDH was used as a loading control. The pERK 1/2 relative to total ERK levels denote BRAF activation status (**B**,**D**); s.e.: short exposure, l.e.: long exposure. Uncropped Western Blots and densitometry analyses can be found at Appendix A. (**E**) Cell proliferation analyses of control (NTC) and Loxl3-silenced (sh2 and sh3) MeL3 cells at different time points after seeding represented as % of cell growth. (**F**) Annexin/PI staining of MeL3 control and Loxl3-silenced cells at the indicated time points after seeding represented as % of total cell population. No comparisons between cells, regardless of apoptosis stage, were significant. (**G**) Representative confocal microscopy images of γH2AX and 53BP1 staining in control NTC and Loxl3-silenced (sh2 and sh3) MeL3 cells. Nuclei were counterstained with DAPI (blue). Scale bars: 20 µm. (**H**) Quantification of γH2AX and 53BP1 foci per nucleus in cells shown in (**G**). A total of 100–150 cells were analyzed per condition (NTC, sh2, and sh3). Mean ± SEM of *n* = 3–5 independent experiments is depicted (**E**,**F**,**H**). The *p*-values were calculated by two-sided unpaired Student’s *t*-test (**E**,**F**,**H**). * *p* < 0.05, ** *p* < 0.01, and *** *p* < 0.001.

**Figure 5 cancers-14-01200-f005:**
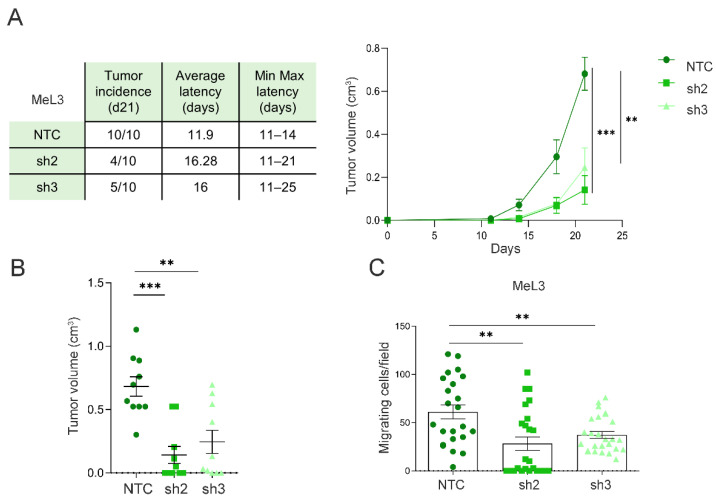
Loxl3 favors melanoma progression. (**A**) Results from three independent tumorigenesis assays of orthotopically injected control (NTC) and Loxl3-silenced (sh2 and sh3) MeL3 cells in immunodeficient mice. The chart (left) includes the tumor incidence at 21 days after injection, average tumor latency, and minimal to maximal latency of *n* = 10 tumors per condition. The graph (right) depicts tumor volume growth up until 21 days after cell injection. (**B**) Volume of tumors originated by control (NTC) and Loxl3-depleted MeL3 cells measured at endpoint (*n* = 10 tumors per condition). (**C**) Number of migrated control (NTC) and Loxl3-silenced (sh2 and sh3) MeL3 cells detected 24 h after seeding. Mean ± SEM of three independent experiments is shown (**A**–**C**). The *p*-values were calculated by two-sided unpaired Student´s *t*-test. ** *p* < 0.01, *** *p* < 0.001.

**Figure 6 cancers-14-01200-f006:**
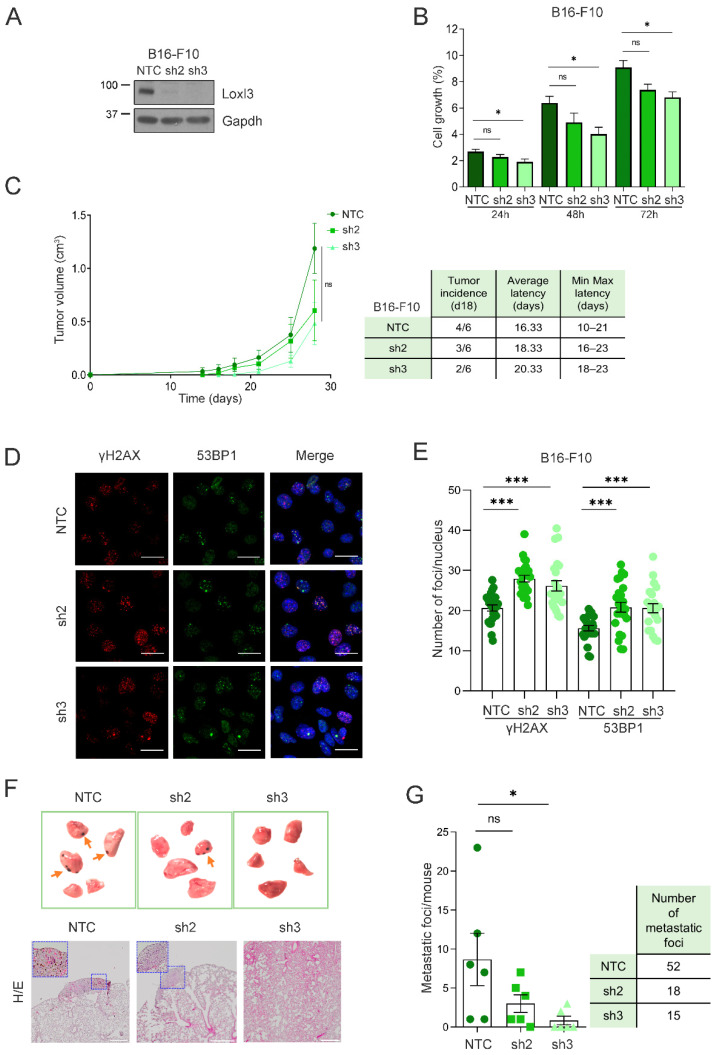
Loxl3 is involved in melanoma metastasis. (**A**) Western blot showing Loxl3 silencing in B16-F10 cell line. Gapdh was used as a loading control. Uncropped Western Blots and densitometry analyses can be found at Appendix A. (**B**) Cell proliferation analyses of control (NTC) and Loxl3-silenced (sh2 and sh3) B16-F10 cells at indicated time points after seeding represented as % of cell growth. Mean ± SEM of *n* = 3 independent experiments is depicted. (**C**) Tumorigenesis assays of orthotopically injected control (NTC) and Loxl3-silenced (sh2 and sh3) B16-F10 cells. The graph (left) depicts tumor volume progression until endpoint (day 28 after cell injection). The chart includes tumor incidence (day 18), average tumor latency, and minimal to maximal latency (*n* = 6 tumors per condition). Mean ± SEM is depicted. (**D**) Representative confocal microscopy images of γH2AX and 53BP1 staining in control NTC and Loxl3-silenced (sh2 and sh3) B16-F10 cells. Nuclei were counterstained with DAPI (blue). Scale bars: 20 µm. (**E**) Quantification of γH2AX and 53BP1 foci per nucleus in cells shown in (**D**). A total of 100–150 cells were analyzed per condition (NTC, sh2, and sh3), and mean ± SEM of *n* = 3 independent experiments is depicted. (**F**) Representative images (top) and tissue H/E staining (bottom) of recovered lungs 15 days after intravenous injection of control NTC and Loxl3-silenced (sh2 and sh3) B16-F10 cells. Orange arrows point to melanocytic metastatic lesions. Scale bars: 500 µm. Blue frames indicate the amplified tissue area shown in the upper left quadrant inside the left and middle panel images. (**G**) Quantification of lung metastatic foci per mice (left) and total number of foci per condition (right) in recovered lungs from experimental metastasis assays detailed in (**F**). Mean ± SEM (*n* = 6 mice per condition) is included. The *p*-values were calculated by two-sided unpaired Student´s *t*-test (**B**,**C**,**E**,**G**). * *p* < 0.05, *** *p* < 0.001; ns: not significant.

**Figure 7 cancers-14-01200-f007:**
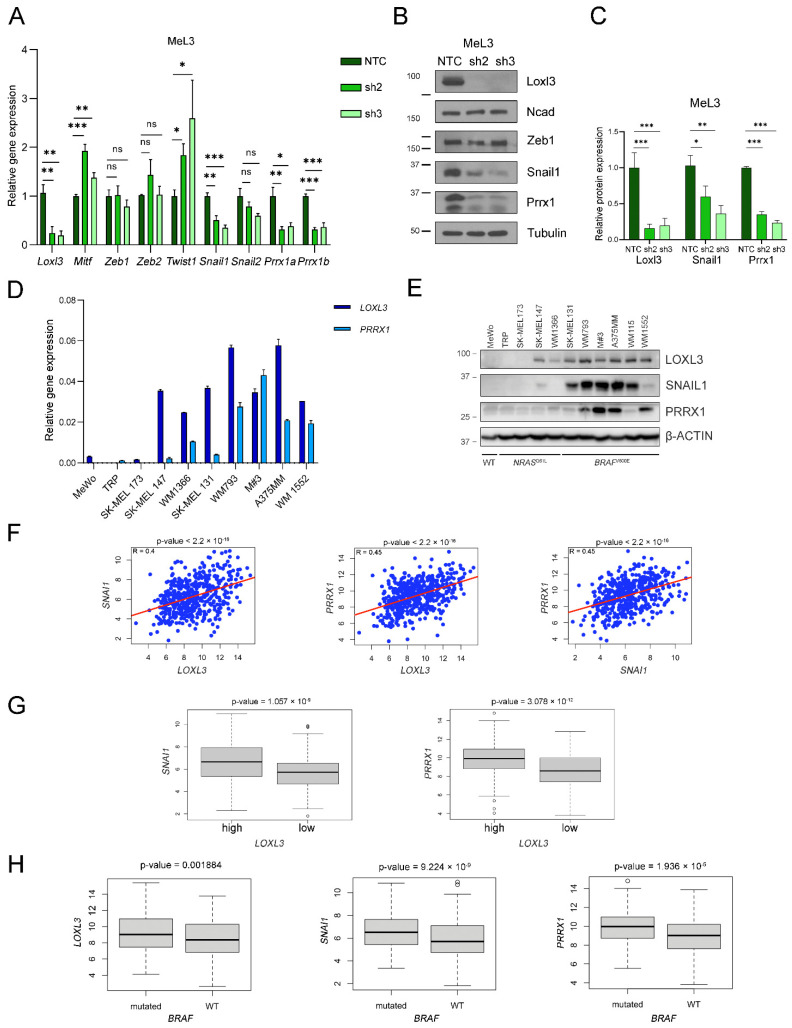
Loxl3 contributes to melanoma cell plasticity. (**A**) Gene expression analyses of *Loxl3*, *Mitf,* and indicated EMT-TFs in control (NTC) and Loxl3-silenced (sh2 and sh3) MeL3 cells relative to internal *Gapdh* mRNA levels and control cells (NTC). (**B**) Immunoblot detection of Loxl3; N-cadherin (Ncad); and Zeb1, Snail1, and Prrx1 EMT-TFs in control (NTC) and Loxl3-silenced (sh2 and sh3) MeL3 cells. Tubulin was used as a loading control. Uncropped Western Blots and densitometry analyses can be found at Appendix A. (**C**) Quantification of Loxl3, Snail1, and Prrx1 protein levels from blots depicted in (**B**) relative to control cells (NTC). Mean ± SEM of *n* = 4–5 independent experiments is shown. The *p*-values were calculated by two-sided unpaired Student´s *t*-test (**A**,**C**). * *p* < 0.05, ** *p* < 0.01, and *** *p* < 0.001; ns: not significant. (**D**) *PRRX1* and *LOXL3* gene expression analyses in a cohort of human melanoma cell lines relative to *L32* mRNA levels. (**E**) Immunoblot analyses of LOXL3, SNAIL1, and PRRX1 in human melanoma cell lines. Cell lines are grouped based on *NRAS^Q61L^*- or *BRAF^V600E^*-prevalent mutations. WT stands for triple WT subtype, which comprises melanomas lacking hot-spot mutations in *BRAF*, *RAS,* or *NF1* genes. Uncropped Western Blots and densitometry analyses can be found at Appendix A. (**F**) Scatter plots of *LOXL3* versus *SNAI1* (left) or *PRRX1* (middle) or *PRRX1* versus *SNAI1* (right) expression for samples in the skin cutaneous melanoma (SKCM) cohort from the TCGA (*n* = 469). The red solid line in each figure represents the regression line. Pearson correlation coefficient (R) with significance (*p*-value) is depicted in each panel. (**G**) Boxplots showing *SNAI1* or *PRRX1* expression plotted against normalized *LOXL3* mRNA levels in human melanoma samples from the TCGA (high expression *n* = 287, low expression *n* = 182). (**H**) Boxplots showing *LOXL3*, *SNAI1,* and *PRRX1* mRNA levels according to *BRAF* mutational status in melanoma samples from the TCGA cohort (*n* = 469).

## Data Availability

The data presented in this study are openly available in TCGA at https://doi.org/10.1016/j.cell.2015.05.044.

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
