# Peer review of "Loxl3 Promotes Melanoma Progression and Dissemination Influencing Cell Plasticity and Survival"

_cancers, 2022, doi:10.3390/cancers14051200_

Round 1

Reviewer 1 Report

The work is a study with two models of melanoma cells for in vitro and in vivo demonstration that Loxl3 contributes to mouse melanogenesis and metastatic dissemination by tumor-autonomous mechanisms. Loxl3 suppression induced decreased melanoma growth and reduced metastatic dissemination. Additional studies in vitro and in vivo confirmed that Loxl3 is important for the progression and metastasis of melanoma by modulating phenotypic switching through Snail1 and Prrx1 EMT transcription factors. Human melanoma cohort of TCGA validated these findings. The results highlight LOXL3 as a therapeutic target in melanoma.

Did the apoptosis analysis (Figure 4F) show differences between NTC and sh2 or NTC and sh3 MeL3 cells if comparing the percentage of early and late apoptosis separately?

Analysis in a large set of human melanoma cases of TCGA is interesting to validate results from mouse cells. Zhang et al. (2021) have shown that high expression of protein LOXL3 correlated with a worse prognosis. How are the expression levels of LOXL3, PRRX1, and SNAI1 in the TCGA melanoma cohort? Is there any difference when comparing cases with BRAF mutation and wild type once it is suggested that LOXL3 cooperates with oncogenic BRAF to promote melanoma transformation and metastasis? Is there any impact in prognostic in overall survival or disease-free survival?

Why the Loxl3 WT tumor-derive Mel3 cells is described as amelanotic (lane 421) if they express TYRP2 (Figure 4B, D)?

Minor points:

The housekeeping gene for qPCR of human melanoma cell lines is described to be ACTB (Figure 7D), but in Material and Methods and in Table S1 it is described to be L32?

Line 234: 37

The work is a study with two models of melanoma cells for in vitro and in vivo demonstration that Loxl3 contributes to mouse melanogenesis and metastatic dissemination by tumor-autonomous mechanisms. Loxl3 suppression induced decreased melanoma growth and reduced metastatic dissemination. Additional studies in vitro and in vivo confirmed that Loxl3 is important for the progression and metastasis of melanoma by modulating phenotypic switching through Snail1 and Prrx1 EMT transcription factors. The human melanoma cohort of TCGA validated these findings. The results highlight LOXL3 as a therapeutic target in melanoma.

Did the apoptosis analysis (Figure 4F) show differences between NTC and sh2 or NTC and sh3 MeL3 cells if analyzing the percentage of early and late apoptosis separately?

Analysis in a large set of human melanoma cases of TCGA is interesting to validate results from mouse cells. Zhang et al. (2021) have shown that high expression of LOXL3 correlated with a worse prognosis. How are the expression levels of LOXL3, PRRX1, and SNAI1 in TCGA melanoma cohort? Is there any difference when comparing cases with BRAF mutation and wild type once it is suggested that LOXL3 cooperates with oncogenic BRAF to promote melanoma transformation and metastasis? Is there any impact in prognostic in overall survival or disease-free survival?

Why is the Loxl3 WT tumor-derived Mel3 cells described as amelanotic (lane 421) if they express TYRP2 (Figure 4B, D)?

Minor points:

The housekeeping gene for qPCR of human melanoma cell lines is described to be ACTB (Figure 7D), but in Material and Methods and in Table S1 it is described to be L32?

Line 234: 37°C, instead of 37ºC.

Line 466: sh2 and sh3, instead of sh3.

Line 267: SNAI1, instead of SNAL1.

Reviewer 2 Report

In this work Vazquez-Naharro et al investigated the role of LOXL3, a known regulator of EMT phenotypic switching, in the development and progression of cutaneous melanoma. For this, they use an elegant combination of conditional mouse models and in vitro systems that allow them to conclude that LOXL3 plays a key role in regulating tumor growth and its metastatic capacity to colonize lymph nodes. Mechanistically, the authors show that LOXL3 might be acting through the regulation of a previously undescribed axis, that involves Snail1 and Prrx1 as EMT-TFs involved in cell plasticity. The manuscript is clearly written, innovative and of clinical relevance, as it proposes LOXL3 targeting as a novel therapeutic strategy in melanoma, where the therapeutic options are currently quite limited (albeit targeting intracellular proteins such as LOXL3 has its limitations, but this is out of the scope of the manuscript).

Although the role that LOXL3 has in repressing cell growth is clear, and the authors have demonstrated this using several in vivo and in vitro approaches, it remains unclear to this reviewer the relative contribution of LOXL3 to the process of metastatic dissemination. Even if the authors show a clear difference in lymph node colonization, specially those distant from primary tumors, in WT vs. LOXL3 KO animals, the step(s) of the metastatic cascade that LOXL3 could be regulating are still unclear. Tail vein injections to study lung colonization, or anchorage-independent growth assays, indicate as well a reduction in metastatic burden, but everything could be still linked to the decrease proliferative abilities of Loxl3 deficient cells. It would be interesting to dissect the involvement of Loxl3 in any of the steps of the metastatic cascade that are different from growth arrest. Alternatively, the authors could possibly center the manuscript in the general effects that Loxl3 has in terms of cell proliferation, both at the primary or metastatic sites.

Although not strictly required, some minor experiments that could strengthen this manuscript, or provide ideas for future projects are:

  • Taking into consideration the role of LOXL3 in collagen crosslinking and general ECM rearrangements, it would be interesting to provide data on how the ECM looks like in WT vs KO animals. This could be even interesting to explain the tumor invasive abilities, if linked for example to the tumor collagen signatures (TACS) described by P. Keely in breast cancer.
  • Do the authors know the proliferative status of cells in the primary tumor or lymph node metastasis? Ki67 stainings would be informative.
  • If available in the databases, it would be interesting to analyze the survival outcomes of patients based on the expression levels of LOXL3, SNAI1 and PRRX1.
  • In some tumor types, genomic instability predisposes tumors to have better chances of responding to immunotherapies. It would be interesting to know if patients with low LOXL3 expression levels (or the combination with SNAI1 and/or PRRX1) (and presumably with higher DNA damage and mutational burden) could display different response patterns to immunotherapies (or other given therapies if the patient cohort is not suitable to address the first one).
